# Measurement and Prediction of Urban Land Traffic Accessibility and Economic Contact Based on GIS: A Case Study of Land Transportation in Shandong Province, China

**DOI:** 10.3390/ijerph192214867

**Published:** 2022-11-11

**Authors:** Zhiguo Shao, Li Zhang, Chuanfeng Han, Lingpeng Meng

**Affiliations:** 1School of Management Engineering, Qingdao University of Technology, Qingdao 266520, China; 2School of Economics and Management, Tongji University, Shanghai 200092, China; 3School of Management Engineering, Shandong Jianzhu University, Jinan 250101, China; 4China (Shanghai) Free Trade Zone Supply Chain Research Institute, Shanghai Maritime University, Shanghai 201306, China

**Keywords:** land traffic accessibility, economic contact, GIS, spatial pattern

## Abstract

As the basic support of regional economic and social development, land transportation is one of the important engines to promote regional development, and its construction and improvement will have an important impact on the regional economic pattern. Based on the road network of Shandong Province, China, in 2020, according to the *Medium and Long-term Development Plan of Comprehensive Transportation Network of Shandong Province (2018–2035)*, this paper uses the GIS network analysis method, weighted average travel time, modified gravity model and other methods to study the land traffic accessibility and economic relation intensity of prefecture-level cities in Shandong Province, China, in 2020 and 2035. The results show that the distribution of land traffic accessibility in Shandong Province, China, shows a certain regional main road pointing characteristic in 2020, and the urban accessibility level gradually decreases along the Beijing–Shanghai high-speed railway and Jinan-Qingdao high-speed railway to the periphery. In 2035, the land traffic accessibility of Shandong Province, China, will be more spatially distributed as “concentric circles”. From 2020 to 2035, the urban land traffic accessibility and the balance of economic contact in Shandong Province, China, will be improved significantly. The research results can provide a theoretical reference for optimizing the traffic network pattern and promoting urban economic contact in Shandong Province, China.

## 1. Introduction

As the carrier of the flow of production factors, the transportation network can connect production and consumption, balance regional supply and demand, attract, multiply and integrate industries, capital and population [1], and provide strong support and guarantee for the steady growth of the regional social economy. At the same time, a perfect transportation network can strengthen the spatial correlation ability of relevant regions [2], accelerate the flow of social and economic factors between cities, and enhance the strength of economic ties between cities greatly [3].

Traffic accessibility is considered a comprehensive index that can evaluate the development status of urban transportation scientifically and effectively by the academic community [4], which can reflect the location, economic development potential and competitiveness of different cities to a certain extent [1]. Hansen [5] first proposed the word “accessibility” and defined it as the interaction opportunity of each node in the transportation network. He applied the gravity method to study the relationship between accessibility and urban land use. Morris [4] defined “accessibility” as the convenience of using a mode of transportation to reach a given location, which has been widely adopted. Ingram [6] defines relative accessibility as the degree to which two locations or points on the same surface are connected. Karst [7] believes that the simplest measure of relative accessibility is the straight line between two points, but infrastructure-based accessibility measures (average travel time, average speed) between two locations can also be a measure of relative accessibility. Shen [8] defines urban space as the whole geographical relationship between residents and the use of transportation, communication and information technology in space. On this basis, he believes that accessibility is a strong indicator to measure this relationship. There are many differences in the academic circle on the precise definition of accessibility. Li Pinghua [9], a Chinese scholar, concluded that the concept of “accessibility” is closely related to the following three aspects: (1) transportation cost (expense, time, etc.); (2) the attraction of nodes (population and economic development status of nodes, etc.); (3) The selection of connection points (based on what purpose to connect to one or more points). At present, the research subjects on traffic accessibility involve comprehensive road networks [10], high-speed rail [11,12], aviation [13,14], highways [15], buses [16], subways [17], etc. The research methods include methods based on space, time, quantity and spatial interaction [18]. Among them, the more common methods are weighted average travel time [19], GIS spatial analysis [20,21] and potential model [22].

Transportation is closely related to the urban economy, and good accessibility in the urban transportation system is closely related to regional spatial integration [23]. Cascetta [24] believes that transportation projects may have an impact on the economy, the location and intensity of activities in a given area, the environment, quality of life and social cohesion. In the book *Regional Development and Its Spatial Structure*, Academician Lu Dadao of China, took Federal Germany as an example to analyze the impact of accessibility changes on regional development and put forward that the level of regional accessibility is a prerequisite for regional development [25]. Mun [26] Mun different traffic network is studied through a numerical simulation model of urban size distribution and family welfare (population and trade patterns between cities, wages and commodity prices, land rent and household utility level), the influence of the population and the results show that the transportation network to promote capital concentration, lead to urban size distribution difference. By calculating the improvement of the national highway network in the United States, Allen [27] concluded that the biggest beneficiaries were concentrated in metropolitan areas and important trade corridors. When verifying the relationship between accessibility and regional economic development, Bartlomiej [28] took the EU transport infrastructure investment from 2004 to 2014 as a positive correlation with regional employment growth. Giuseppe [29] developed a model of the interaction between Land Use and Transport Interaction (LUTI) by analyzing residents’ intention to choose the location of their households and validated it with a real case study. Francesco [30], from the theoretical background to the theoretical evolution of a kind of social economic model currently expressed, proposed the Spatial Economic Transport Interaction (SETI) from the national and urban scale, in order to capture the two-way relationship between the space economy and the transportation system. With the construction and improvement of rapid transit infrastructure, research on the connection between transportation accessibility and the urban economy is gradually enriched. In terms of research content, scholars mainly focus on the division and accessibility of urban hinterland [31], the spatiotemporal coupling characteristics of accessibility and urban economy [32], and the characteristics of accessibility and economic diffusion [33], etc. In terms of research scale, they pay more attention to the relationship between urban traffic accessibility and economic contact at the spatial scale of urban agglomerations or metropolitan areas [34]. For example, Matsumoto [35] analyzes the pattern of air passenger and cargo flows within and between Asia, Europe and the Americas from the perspective of international air traffic flows based on the gravity model. T. Bunnell [36] analyzed the spatial evolution process of the Kuala Lumpur metropolitan area by analyzing the changes related to transportation, housing and information technology within the metropolitan area. Kiyoshi [37] proposed a dynamic multi-regional growth model with a free flow of capital and population, emphasized the impact of geographical differences in the high-speed rail system on regional economic development, and simulated the dynamic process of urban system economic development. Wang Hui et al. [38] used the weighted average travel time and gravity model to analyze the impact of high-speed rail construction on accessibility and economic ties in the Beijing–Tianjin–Hebei region and believed that the effect of high-speed rail on improving the economic potential of core cities along the route was significantly higher than that of peripheral cities. Allen [27] calculated that the greatest beneficiaries of improvements to the US national highway network would be concentrated in metropolitan areas and important trade corridors. When verifying the relationship between accessibility and regional economic development, Bartlomiej took the positive correlation between transportation infrastructure investment and regional employment growth in the EU from 2004 to 2014 [28].

To sum up, there are abundant research results on the connection between traffic accessibility and economy, which mainly focus on the evolution of regional transport accessibility and economic connection, the impact of different transport mode changes on regional economy and accessibility, and the impact of the establishment of a traffic channel on regional accessibility and economy [39]. However, there is still a lack of studies on the relationship between urban traffic accessibility and economic contact at the provincial spatial scale. In addition, existing studies mainly measure and analyze the existing and historical road network, but lack the measurement and analysis of the planned road network on the future traffic accessibility and economic connection of the city. In view of this, this article calculates the land traffic accessibility and economic relation intensity of each prefecture-level city in Shandong Province, China, in 2020 and 2035, and analyzes the current and future urban land traffic accessibility, economic relation intensity level and spatial distribution in Shandong Province, China, based on the road network of Shandong Province, China, in 2020 and 2035. It aims to optimize and improve the pattern of the transportation network in Shandong Province, China, promote the rational allocation of transportation resources in Shandong Province, China, strengthen urban economic ties and cooperation, and provide theoretical support for the formulation of development strategy in Shandong Province, China.

The rest of this study is arranged as follows. In Section 2, the introduction of the study area, the data used for empirical research, and the methods employed in this study are elaborated. In Section 3, the spatial and temporal patterns of urban land transport accessibility and economic linkages in Shandong Province, China, in 2020 and 2035 were analyzed. Section 4 includes a brief summary, and expounds the significance of urban planning and the future research direction.

## 2. Materials and Methods

### 2.1. Survey Region

Located in the eastern part of the Chinese mainland and in the lower reaches of the Yellow River, Shandong Province is rich in natural resources and has a complete industrial system, which is an important strategic resource base of China. Additionally, the transportation network in Shandong Province extends in all directions, with Beijing–Shanghai, Beijing–Kowloon and other railways and highway networks all over the province, which is an important transportation hub connecting the North and South of China. By the end of 2020, Shandong Province had opened 2110 km of high-speed railways and 286,800 km of highways, an increase of 6489 km over the previous year. In the *Medium and Long-term Development Plan of Comprehensive Transportation Network of Shandong Province (2018–2035)* [40], By 2035, Shandong Province plans to form a comprehensive transport corridor of “four horizontal and five vertical” to effectively connect with China’s “ten vertical and ten horizontal” comprehensive transport corridor. The land transportation network of Shandong Province, China, will undergo significant changes in the future.

### 2.2. Data Sources and Processing

In this paper, vector data of administrative divisions, expressways, national roads and provincial roads of Shandong Province, China, in 2020, were obtained through the Shandong Provincial Geographic Information Public Service platform for data visualization and processing. The railway data were derived from a 1:250,000 national basic geographic information database. The relevant elements were projected to WGS_1984_World_Mercator to obtain the road network of Shandong Province, China, in 2020, as shown in Figure 1. The planned road network data for 2035 were derived from the *Medium and Long-term Development Plan of Comprehensive Transportation Network of Shandong Province (2018–2035)* (hereinafter referred to as the *Plan*). The vector data of the planned road network were obtained by vectorization after sampling and geo-registration of the planning pictures by ArcGIS. Due to the availability and limitation of the planned road network, this study only involves the planning data of high-speed railways and expressways, and other road network data were replaced by the data of 2020. After combining relevant elements, the planned road network of Shandong Province, China, in 2035 can be obtained, as shown in Figure 2. The socio-economic data of the 16 prefecture-level cities in Shandong Province, China, involved in this paper were all derived from Shandong Statistical Yearbook. The socio-economic data of 2035 were based on 2020 and calculated by the annual average growth rate of the socio-economic data from 2005 to 2020.

### 2.3. Methods

#### 2.3.1. Framework of Methodology

The overall theme of this paper is to explore and predict the spatial distribution of urban land transport accessibility and urban economic linkages. The method framework is shown in Figure 3.

#### 2.3.2. Traffic Accessibility Measurement

Traffic accessibility can reflect the difficulty of transportation between regions [19], and weighted average travel time is the main method to measure regional traffic accessibility. This method takes into account the influence of node city development level on traffic accessibility. Its calculation process is intuitive and simple, and the measurement results are representative [1]. Therefore, this paper uses weighted average travel time to calculate the land traffic accessibility level of each city in Shandong Province, China. The equation is as follows:(1)$$A_{i}=\frac{\sum_{j=1}^{n}\left(T_{ij}\times M_{j}\right)}{\sum_{j=1}^{n}M_{j}}$$
(2)$$M_{j}=\sqrt{G_{j}\times P_{j}}$$
where $A_{i}$ denotes the accessibility of nodes in the region. The smaller $A_{i}$ is, the better the urban accessibility; otherwise, the worse the urban accessibility. $T_{ij}$ is the time cost distance from the city i to the city j by land transportation; M is the flow of a certain social and economic factor flow of a city and activity destination within the scope of the evaluation system, that is, it represents the economic strength of the economic center or its radiation force or attraction to the surrounding areas. Generally, it is represented by the comprehensive value of the two indexes of the gross regional product (100 million yuan) and the urban permanent resident population (10,000 people). $G_{j}$ represents the gross regional product of node cities, and $P_{j}$ represents the resident population of node cities. n is the total number of nodes in the evaluation system except ground nodes.

The time cost distance among prefecture-level cities was calculated based on ArcGIS 10.8.0.12790 software. Firstly, according to the *Highway Engineering Technical Standards (JTG B01–2014)* and *High-Speed Railway Design Code (TB10621–2014)*, the average running speeds of different types of roads are obtained, as shown in Table 1. Then, different types of road elements are obtained through the map vectorization method (see Figure 1 and Figure 2). According to the content of Table 1, corresponding fields are added to Figure 1 and Figure 2 in ArcGIS and the corresponding values are calculated to obtain the cost of passing through each section of the road. Finally, taking the location of municipal governments at different levels as the starting point, the O-D matrix method in the Network Analyst module of ArcGIS is used to calculate the shortest time distance $T_{ij}$ from city i to city j under the land-integrated transportation route.

#### 2.3.3. Modified Gravity Model

Using the shortest time of inter-city land transportation to modify the gravity model, the economic contact between cities in Shandong Province, China, is measured. The equation is as follows:(3)$$R_{ij}=\frac{\sqrt{P_{i}G_{i}}\sqrt{P_{j}G_{j}}}{T_{ij}^{d}}$$
(4)$$R_{i}=\sum_{j=1}^{n}R_{ij}$$
where $R_{ij}$ is the strength of the economic connection between cities i and j, and the larger the value, the closer the economic connection. $R_{i}$ represents the strength of external economic ties of city i, reflecting the level of economic ties between the city and other cities. $P_{i}$ and $P_{j}$ are the population scales of node cities i and j; $G_{i}$ and $G_{j}$ are the gross national product (GDP) of node cities i and j; $T_{ij}$ is the shortest land transportation time between the city i and the city j; d is the friction coefficient, which reflects the decay rate of influence of node city, and its empirical value is 2 [41].

#### 2.3.4. Equilibrium Measurement

The coefficient of variation was used to verify whether the planned road network would enhance or narrow the gap in accessibility or intensity of economic connection between cities [42]. According to Equations (1) and (4), the land traffic accessibility and economic contact intensity of each prefecture-level city in Shandong Province, China, in 2020 and 2035 were obtained. The coefficient of variation method was used to calculate the coefficient of variation in land transportation accessibility and the intensity of economic ties with foreign countries of each prefecture-level city from 2020 to 2035, and the balance between the differences in land traffic accessibility and the economic contact under the existing road network and the planned road network was analyzed. The equation is as follows:(5)$$CV=(\frac{SD}{Mean})\times 100\%$$
where CV represents the coefficient of variation; SD represents the standard deviation of urban land traffic accessibility or economic contact; Mean represents the average of urban land traffic accessibility or the economic contact.

## 3. Results

### 3.1. Land Traffic Accessibility Measurement and Prediction Analysis

ArcGIS network analysis method was used to calculate the shortest time cost distance between each node city. At the same time, population and economic dual indexes were taken as weights, and the results were obtained by using Equations (1) and (2). Then, the inverse distance interpolation method and natural discontinuity point classification method were used to obtain the spatial distribution of land traffic accessibility in Shandong Province, China, in 2020, and the prediction of the spatial distribution of land transportation accessibility in Shandong Province in 2035, as shown in Figure 4 and Figure 5. It can be seen from Figure 4 and Figure 5 that the spatial distribution of urban land transportation accessibility in Shandong Province, China, shows a “central–periphery” pattern with the provincial capital city as the core and gradually increasing to the peripheral areas. Compared with the central region, the northeast and southwest regions of Shandong Province, China, have poor urban land traffic accessibility, which is the weak area of urban land transportation accessibility in Shandong Province, while the central region of Shandong Province is the advantageous area of urban land traffic accessibility. In 2020, the distribution of land transportation accessibility in Shandong Province will show certain characteristics of regional main roads, and the urban accessibility level will gradually decrease to the periphery along the Beijing–Shanghai high-speed railway and the Jiji-Qingdao high-speed railway. The construction of a high-speed railway can narrow the transportation accessibility gap between the outer cities (such as Dezhou and Zaozhuang) and the central cities (such as Jinan, Taian and Zibo) in Shandong Province. In 2035, the spatial distribution of land traffic accessibility in Shandong Province, China, has no obvious road direction characteristics. It shows that by 2035, Shandong Province will have formed a relatively complete land transportation network, with fast passageways all over the province, which will significantly improve the traffic accessibility level of cities in Shandong Province, China, and the accessibility level will be more spatially distributed in “concentric circles”. For cities located in areas with weak land transportation accessibility, such as Yantai, Weihai and Heze, it is difficult to bridge the transportation accessibility gap with central areas even if they build high-speed railways, expressways and other fast transportation arteries. However, these cities can compensate for the gap in land transport by expanding their transport links with the outside world through air and water transport.

In order to further analyze the change difference in land traffic accessibility in Shandong Province, China, in 2020 and 2035, the spatial difference in land traffic accessibility change rate in Shandong Province, China, (2020–2035) was obtained from the variability analysis of urban land transport accessibility before and after the planning, as shown in Figure 6. Figure 6 shows that after the planned road network is completed in 2035, the overall land traffic accessibility of Shandong Province, China, will change significantly, and the change rate of land traffic accessibility of 16 prefecture-level cities will all exceed 20%. Among them, the central area of Shandong Province, China, has a small change, while the marginal areas have a large change, such as Weihai, Yantai, Dongying, Liaocheng, Heze, Linyi, Rizhao, which are located in the peripheral areas of Shandong Province, China, the change rate is more than 40%. It can be seen that road network planning can improve the level of land traffic accessibility greatly in Shandong Province, China, and further maintain the advantage of traffic accessibility in the central region, and narrow the accessibility difference between the marginal and central regions. By comparing Figure 4 and Figure 6, it can be found that the spatial distribution of land transportation accessibility in 2020 is similar to the spatial distribution pattern of the horizontal change rate of accessibility before and after the planned road network. This shows that on the basis of the existing road network, the *Plan* is to further optimize and supplement the transportation network in the areas with inconvenient transportation and poor transportation accessibility so that the land transportation fast track (such as high-speed rail and expressway) network can be spread throughout the province. It can be seen that by 2035, the completion and improvement in the planned road network can further solve the problems of the existing road network layout in Shandong Province, China, is not dense enough, the traffic connection is not tight enough, the urban traffic is not smooth enough, and the traffic accessibility in the marginal areas is poor. Thus, Shandong Province can realize the goal of “focusing on key points, reinforcing weak points, strong and weak items, and further optimizing and improving the comprehensive transportation network layout of the province” [40].

### 3.2. Measure and Forecast Analysis of Economic Relation Intensity

According to Formula (3), the economic contact among prefecture-level cities in Shandong Province, China, is calculated, and the XY to line point-to-line tool in ArcGIS is used to draw the urban economic relation intensity in Shandong Province, China, in 2020 and 2035, as shown in Figure 7 and Figure 8. In addition, the total urban economic relation intensity is summarized to obtain the total urban economic relation intensity in Shandong Province, China, in 2020 and 2035, as shown in Table 2. It can be seen from Figure 7 that the strength of economic contact between cities in the central region of Shandong Province, China, is strong, among which the strength of economic contact between Jinan and Zibo, and between Zibo and Weifang both exceed 10 million yuan. The economic relation intensity between Jinan and Zibo, Tai’an, Weifang, and Qingdao and Weifang all exceed 5 million; The strength of economic contact between cities in the western and northern regions of Shandong Province, China, is weak, and most of the economic relation intensity between cities are below 1 million, that is because the western and northern parts of Shandong Province, China, are short of rapid traffic arteries, and the social and economic foundation of some cities is relatively weak, so the traffic accessibility is poor compared with the central part of Shandong Province, China. The economic contact of cities in Shandong Province, China, shows a pattern of “strong in the middle and weak in the periphery”, and the core edge features are significant. As can be seen from Figure 8, it can be predicted that by 2035, the level of urban economic connections and networking in Shandong Province, China, will be improved greatly, the central radiation role will be strengthened, and the marginalization will be weakened, a total of 39 groups of cities have more than 10 million economic ties, forming an urban economic connection network with the inland city of Jinan as the core and the coastal city of Qingdao as the core.

According to Table 2, the total pattern of urban economic relation intensity in Shandong Province, China, changed greatly from 2020 to 2035. Among them, Qingdao, Heze and Liaocheng made great progress in the ranking of the total economic relation intensity. This is because, by 2035, Shandong Province will fully form “four horizontal and five vertical” comprehensive transportation channels, the land traffic accessibility of the marginal areas can be improved greatly, the inter-city transportation time will be shortened, so the total economic relation intensity of cities located in the periphery of Shandong Province, China, will be improved greatly. In 2020, the total economic relation intensity between Dezhou and Zaozhuang ranked moderately high in the whole province, while in 2035, the total economic relation intensity between Dezhou and Zaozhuang dropped more. This is because the Beijing–Shanghai high-speed railway runs through Shandong Province and passes through Dezhou and Zaozhuang, which greatly facilitates transportation between them. Therefore, the land traffic accessibility between Dezhou and Zaozhuang is better than that of other marginal areas, and the strength of the economic connection is also relatively good. However, with the completion and improvement of the planned road network, the land transportation accessibility of other cities in Shandong Province, China, is also improved, while the advantages of traffic accessibility of Dezhou City and Zaozhuang City are decreased. In addition, the social and economic foundation of Dezhou and Zaozhuang is weaker than that of Linyi, Yantai, Heze and other peripheral cities, so the total amount of economic relation intensity between Dezhou and Zaozhuang ranked lower in 2035.

### 3.3. Distribution Equilibrium Analysis of Accessibility and Economic Contact

According to Formula (5), the coefficient of variation in urban land traffic accessibility and economic relation intensity in Shandong Province, China, is calculated, as shown in Table 3, so as to analyze the balance of differences between urban land traffic accessibility and economic relation intensity in Shandong Province, China, before and after the completion of the planned road network. According to Table 3, after the completion of the planned road network, the value of urban land traffic accessibility in Shandong Province, China, is reduced to 59.65% on average, and the coefficient of variation is reduced to 18.31%. The overall land traffic accessibility in Shandong Province, China, significantly improved, the difference in land traffic accessibility between cities became smaller, and the balance of land traffic accessibility improved. In addition, after the completion of the planned road network, the urban economic relation intensity in Shandong Province, China, expanded to 10.60 times on average, and the coefficient of variation decreased to 55.50%, which indicated that the construction of the planned road network could improve the intensity of economic contact between cities and improve the equilibrium of economic contact between cities. By comparing the coefficient of variation between urban land traffic accessibility and economic relation intensity in Shandong Province, China, the difference in urban economic relation intensity is more significant than that of urban land traffic accessibility.

## 4. Discussion

This study analyzes the accessibility of urban land transportation, the level of economic links and their spatial distribution in Shandong Province in 2020 and 2035, and finds that the level of land transportation in Shandong Province, China, will be greatly improved by 2035. The accessibility level of land transportation in Shandong Province presents a “concentric circle” distribution on the whole, but it also shows a certain regional main road orientation characteristic. For example, in 2020, the level of urban accessibility will gradually decrease to the periphery along the Beijing–Shanghai high-speed railway and the Beijing–Qingdao high-speed railway. Therefore, the construction of high-speed railways can narrow the traffic accessibility gap between the peripheral cities (Dezhou, Zaozhuang, etc.) and the central cities (Jinan, Taian, Zibo, etc.) in Shandong Province. The construction and development of high-speed railways can affect regional transport accessibility, shorten people’s travel time, and make people travel more convenient and comfortable. This coincides with the concept of “Mobility as a Service” (MaaS). The development of modern transportation technology provides people with better services (shorter time, appropriate price and comfortable journey) for long-distance travel, which makes people more inclined to public transportation [43]. As a result of the gradual penetration of emerging technologies, the transformation of transport “production” is underway, resulting in lower unit costs of transport supply and increased benefits for transport users, enabling sustainable development as defined by the three pillars of economic, social and environmental development. Thus, MaaS consists of three main elements: customer-centric supply design, sustainability goals, and (emerging) technology adoption [44]. The paper predicts that land transportation accessibility in Shandong Province will be greatly improved by 2035 compared with that of 2020. However, due to the geographical location outside Shandong Province, the land transportation accessibility level of Heze, Yantai and Weihai is relatively low. According to the concept of MaaS, it is suggested that Shandong Provincial government can broaden its communication channels with the outside world by increasing the means of transportation supply in the subsequent planning, so as to make up the gap in land transportation. For example, by strengthening the construction of land transport infrastructure in these cities, we can also build transport infrastructure such as airports and ports to provide residents with multi-mode and multi-service travel modes. In addition, it is suggested to give full play to transportation advantages in areas with superior land transportation accessibility such as Jinan, Zibo, Tai’an and other cities in Shandong Province, establish transportation hubs, and vigorously develop industries that rely on land transportation to promote local economic development. It will strengthen the fulcrum and node roles of coastal port cities such as Qingdao, Yantai, Rizhao and Weihai and inland central cities such as Jinan and Zibo, and deepen cooperation on infrastructure connectivity, trade, industrial investment, energy and resources, and people-to-people exchanges among countries along the Belt and Road. Based on the needs of residents’ travel and local economic development, through the construction of a unified digital platform for comprehensive transport services [45], the connectivity of multi-level transport modes is achieved, a transport service chain is established, and a customer-centered transport network is established to provide customers with multi-mode and multi-service travel modes.

This study has made a spatiotemporal analysis of urban land transportation accessibility and economic connection intensity in Shandong Province, China, in 2020 and 2035, and has a macro understanding of the traffic distribution pattern in Shandong Province, China. However, the study did not involve the road delay index, traffic transfer waiting time, road congestion time, traffic costs and other factors, so the results of traffic accessibility tend to be theoretical and idealized, which will provide a direction for subsequent refined research. In addition, the transportation network contains a variety of transportation modes, and the land transportation level cannot represent the overall traffic level of the city. In the future, a comprehensive study on urban traffic accessibility will be carried out by combining land, water, air and other transportation modes.

## 5. Conclusions

Based on the road network data of Shandong Province, China, in 2020, we calculated the land traffic accessibility and economic relation intensity of all prefecture-level cities in Shandong Province, China, in 2020 and 2035 according to the road network planning, and analyzed the current and future urban land traffic accessibility, economic link intensity level and spatial distribution pattern of Shandong Province, China. The main conclusions are as follows:(1)The spatial distribution of land traffic accessibility in Shandong Province, China, showed a “central–periphery” pattern with the provincial capital city as the core and gradually increasing to the peripheral areas. In 2020, the distribution of land traffic accessibility in Shandong Province, China, showed a certain regional main road pointing characteristic, and the urban accessibility level decreased gradually along the Beijing–Shanghai Railway and Ji–Qing Railway to the periphery. In 2035, the spatial distribution of land traffic accessibility in Shandong Province, China, has no obvious road direction characteristics, and the accessibility level tends to be more “concentric circles” in space. From 2020 to 2035, the overall land traffic accessibility in Shandong Province, China, will change significantly, with the change rate of land traffic accessibility in 16 prefecture-level cities all exceeding 20%. Among them, the central area of Shandong Province, China, changes less, whereas the marginal area changes more. The spatial distribution of land traffic accessibility in 2020 is similar to the spatial distribution pattern of the horizontal change rate of accessibility before and after the planned road network. It shows that the planned road network greatly improves the accessibility of land transportation in Shandong Province, China, further maintains the advantage of transportation accessibility in the central areas, and reduces the difference in accessibility between the peripheral area and the central area. By 2035, with the completion of the planned road network, the land traffic accessibility of cities located in the peripheral areas of Shandong Province, China, will be improved significantly. This will bring new opportunities and development to the industries that depend on land transportation in the peripheral cities of Shandong Province, China, and then bring local economic vitality and promote urban construction and social development.(2)In 2020, the economic contact of cities in Shandong Province, China, showed a pattern of “strong in the middle and weak in the periphery”, and the core edge features were significant. Among them, the strength of economic contact between Jinan and Zibo was the highest, followed by that between Zibo and Weifang. The strength of economic contact between cities in central Shandong Province, China, was about 10 times that between cities in western and northern Shandong Province. This is because the western and northern regions of Shandong Province, China, are short of rapid traffic arteries, and the social and economic foundation of some cities is relatively weak, so traffic accessibility is relatively poor compared with the central region of Shandong Province, China. By 2035, the network level of urban economic contact in Shandong Province, China, will be improved greatly, the central radiation role will be strengthened, and the marginalization will be weakened. Overall, a network of urban economic contact with Jinan and Qingdao as the core will be formed.(3)After the completion of the planned road network, the urban land traffic accessibility in Shandong Province, China, decreased to 59.65% on average, the coefficient of variation decreased to 18.31%, the urban economic relation intensity expanded to 10.60 times the original, the coefficient of variation decreased to 55.50%. This shows that the completion of the planned road network will make the urban land traffic accessibility and the balance of economic contact in Shandong Province, China, obviously improved.

## Figures and Tables

**Figure 1 ijerph-19-14867-f001:**
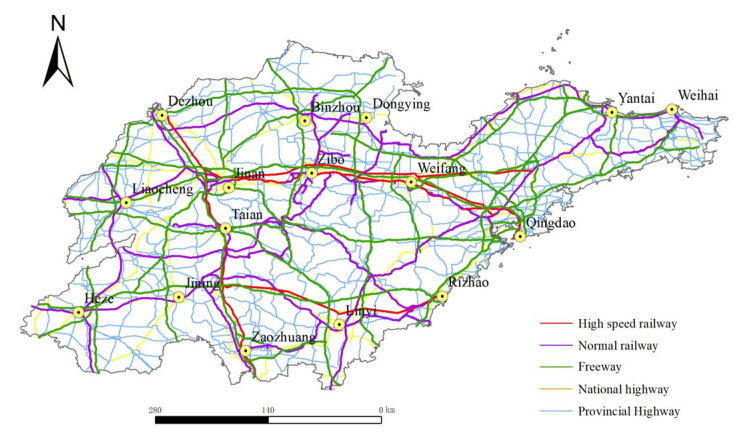
Road network of Shandong Province, China, in 2020.

**Figure 2 ijerph-19-14867-f002:**
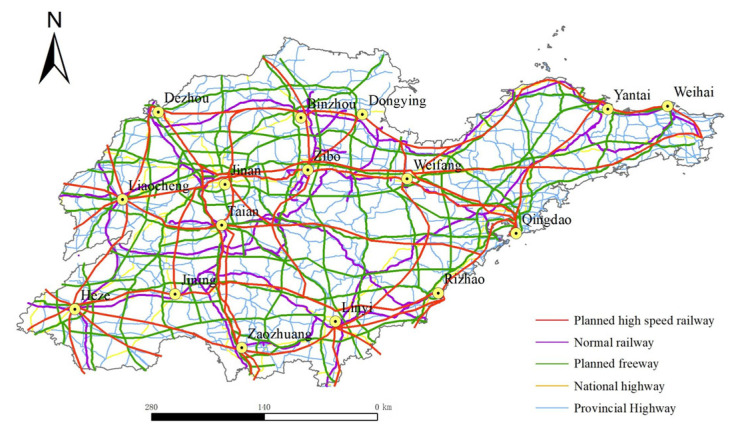
Planned road network of Shandong Province, China, in 2035.

**Figure 3 ijerph-19-14867-f003:**
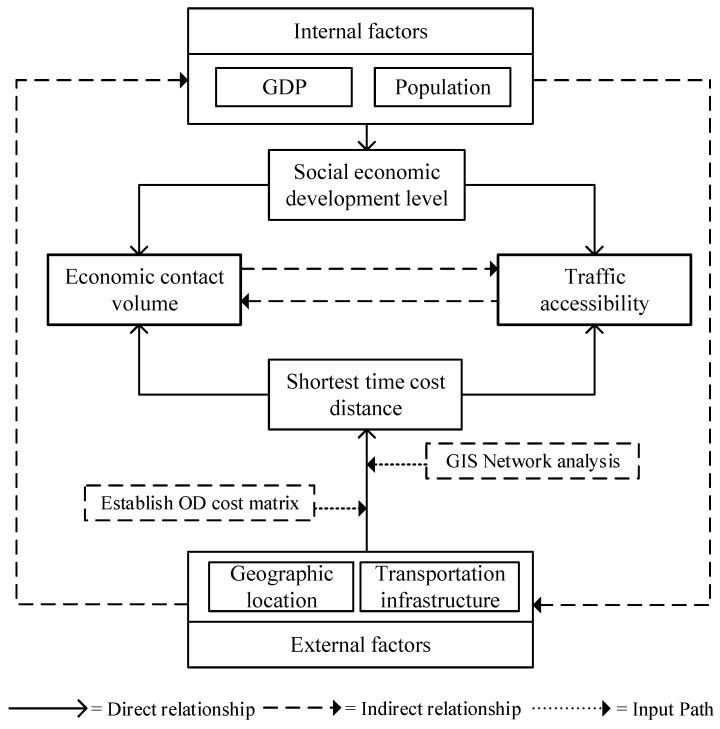
Framework of Methodology.

**Figure 4 ijerph-19-14867-f004:**
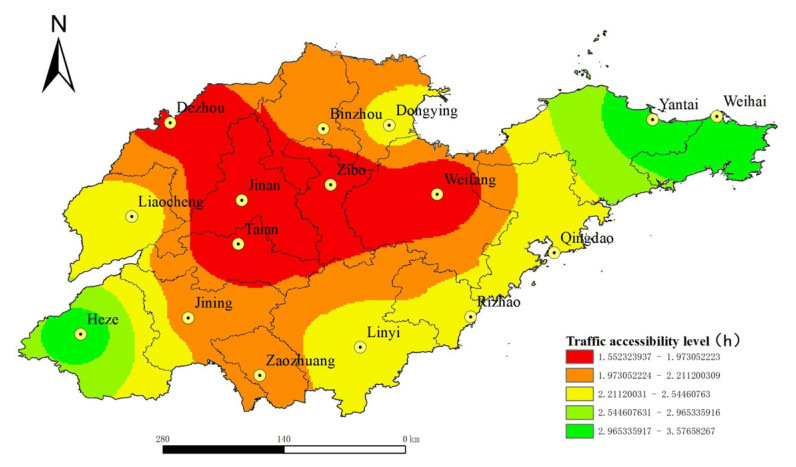
Spatial distribution of land traffic accessibility in Shandong Province, China, in 2020.

**Figure 5 ijerph-19-14867-f005:**
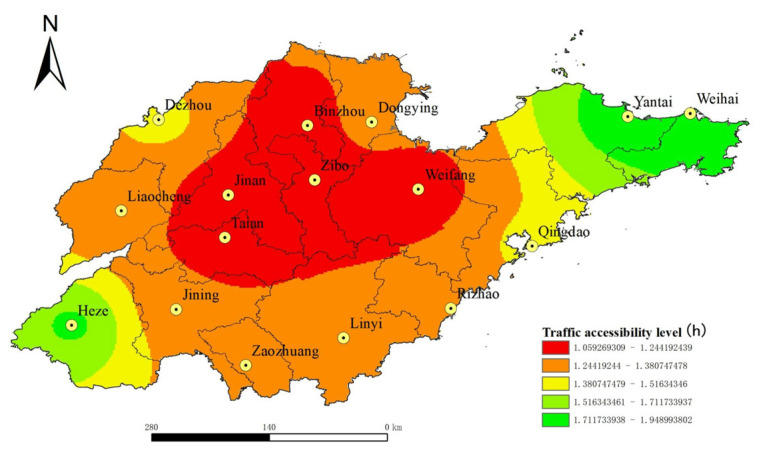
Prediction of spatial distribution of land traffic accessibility in Shandong Province, China, in 2035.

**Figure 6 ijerph-19-14867-f006:**
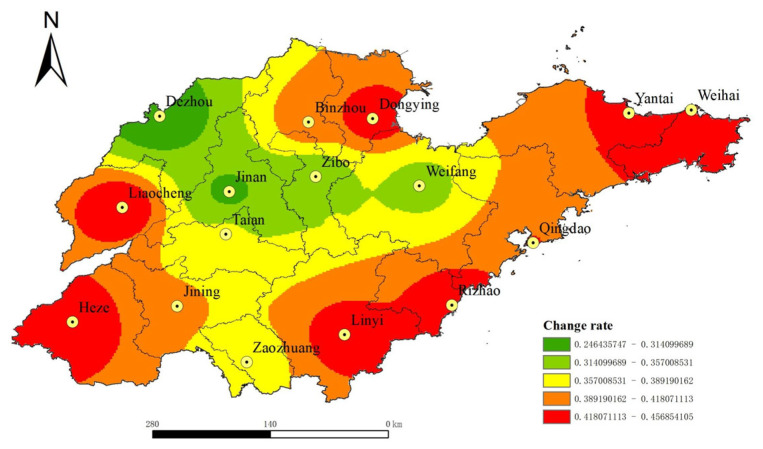
Spatial difference in land traffic accessibility change rate in Shandong Province, China (2020–2035).

**Figure 7 ijerph-19-14867-f007:**
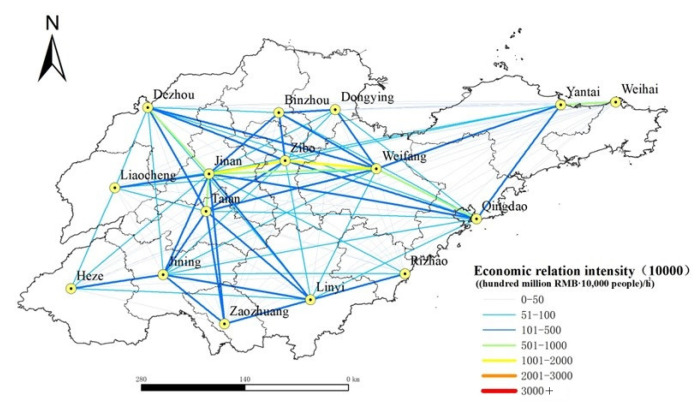
Urban economic relation intensity in Shandong Province, China, in 2020.

**Figure 8 ijerph-19-14867-f008:**
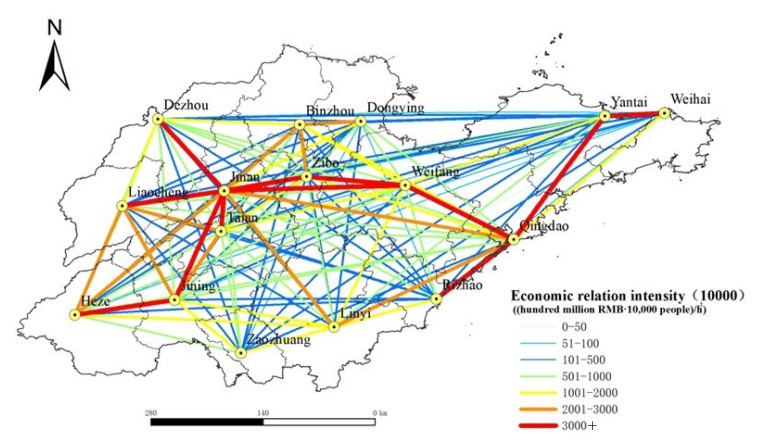
Prediction of urban economic relation intensity in Shandong Province, China, in 2035.

**Table 1 ijerph-19-14867-t001:** Average running speed of different types of roads.

Road Type	High-Speed Railway	Normal Railway	Freeway	National Highway	Provincial Highway
speed (km·h^−1^)	300	120	100	80	60

**Table 2 ijerph-19-14867-t002:** Total urban economic relation intensity in Shandong Province, China, in 2020 and 2035.

City	Total Economic Relation Intensity in 2020 ((Hundred Million RMB·10,000 People)/h^2^)	Rank	Total Economic Relation Intensity in 2035 ((Hundred Million RMB·10,000 People)/h^2^)	Rank
Jinan	49,879,693.08	1	461,229,665.6	1
Qingdao	21,913,587.02	5	313,107,826.3	2
Weifang	37,151,704.75	2	281,939,545.4	3
Taian	22,191,904.62	4	244,154,143	4
Jining	17,015,714.54	6	231,013,541	5
Zibo	34,348,646.4	3	194,201,918	6
Heze	6,897,865.831	13	180,953,069.8	7
Linyi	15,310,512.72	8	163,193,541.4	8
Yantai	10,734,351.04	9	151,988,600.1	9
Liaocheng	7,020,214.285	12	148,289,389.3	10
Binzhou	10,203,135.06	10	140,475,056.2	11
Dezhou	16,388,704.26	7	113,083,443.6	12
Rizhao	6,582,575.236	14	85,375,883.05	13
Dongying	5,885,325.977	16	76,628,511.46	14
Weihai	6,117,984.57	15	72,735,838.29	15
Zaozhuang	8,627,425.902	11	70,702,513.5	16

**Table 3 ijerph-19-14867-t003:** Coefficient of variation between urban land traffic accessibility and economic relation intensity in Shandong Province, China.

	Land Traffic Accessibility in 2020	Land Traffic Accessibility in 2035	Economic Relation Intensity in 2020	Economic Relation Intensity in 2035
SD	0.56924997	0.251887101	12,621,933.28	101,603,812.30
Mean	2.305626332	1.375354588	17,266,834.08	183,067,030.37
CV	24.69%	18.31%	73.10%	55.50%

## Data Availability

The data presented in this study are available on request from the corresponding author.

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
