# Peer review of "Measurement and Prediction of Urban Land Traffic Accessibility and Economic Contact Based on GIS: A Case Study of Land Transportation in Shandong Province, China"

_ijerph, 2022, doi:10.3390/ijerph192214867_

Round 1

Reviewer 1 Report

The paper proposes a method for predicting of urban land traffic accessibility and economic contact based on GIS. The issue is relevant and the paper seem interesting. Despite its strengths, the current version presents some limits. In the following some broad and specific comments.

Reviewer 2 Report

The authors present a comprehensive study of land transportation volumes in Shandong province, China.

A thorough literature review forms the basis for the article and the study's relevance is adequately conveyed.

The basic methodology and framework presented are well-prepared and documented.

The results provide added value and insights for transportation infrastructure planning (in Shandong province, China) and research.

I recommend publication of the article but would suggest careful editing as some typos stand out. Missing spaces after commas (e.g., in lines 44, or 281).

In addition, I would adjust the Figure 5 caption to speak of a forecast or prediction.

Round 2

Reviewer 1 Report

The new version of the paper overcomes the previous limits.

The author consider my comments and include revisions in the new version.

My last comment concerns  the concept of Sustainable Mobility as a Service (S-MaaS). The authors refer to the MaaS concept with anew reference. I suggest the authors to refer to the recent works published in "Information 2022, 13". You can focus on "Supply Analysis". 
